# Anti-Müllerian Hormone Levels among Female Firefighters

**DOI:** 10.3390/ijerph19105981

**Published:** 2022-05-14

**Authors:** Samantha Davidson, Sara Jahnke, Alesia M. Jung, Jefferey L. Burgess, Elizabeth T. Jacobs, Dean Billheimer, Leslie V. Farland

**Affiliations:** 1Department of Epidemiology and Biostatistics, Mel and Enid Zuckerman College of Public Health, University of Arizona, Tucson, AZ 85724, USA; sdavidson@email.arizona.edu (S.D.); ajung1@email.arizona.edu (A.M.J.); jacobse@arizona.edu (E.T.J.); dean.billheimer@arizona.edu (D.B.); 2Center for Fire, Rescue & EMS Health Research, NDRI-USA, Inc., Leawood, KS 66224, USA; jahnke@ndri-usa.org; 3Department of Community, Environment & Policy, Mel and Enid Zuckerman College of Public Health, University of Arizona, Tucson, AZ 85724, USA; jburgess@arizona.edu; 4Cancer Prevention and Control Program, University of Arizona Cancer Center, Tucson, AZ 85719, USA; 5Department of Obstetrics and Gynecology, College of Medicine-Tucson, University of Arizona, Tucson, AZ 85724, USA

**Keywords:** anti-müllerian hormone, firefighter health, occupational exposures, reproductive health

## Abstract

Female firefighters have occupational exposures which may negatively impact their reproductive health. Anti-müllerian hormone (AMH) is a clinical marker of ovarian reserve. We investigated whether AMH levels differed in female firefighters compared to non-firefighters and whether there was a dose-dependent relationship between years of firefighting and AMH levels. Female firefighters from a pre-existing cohort completed a cross-sectional survey regarding their occupational and health history and were asked to recruit a non-firefighter friend or relative. All participants provided a dried blood spot (DBS) for AMH analysis. Linear regression was used to assess the relationship between firefighting status and AMH levels. Among firefighters, the influence of firefighting-related exposures was evaluated. Firefighters (*n* = 106) and non-firefighters (*n* = 58) had similar age and BMI. Firefighters had a lower mean AMH compared to non-firefighters (2.93 ng/mL vs. 4.37 ng/mL). In multivariable adjusted models, firefighters had a 33% lower AMH value than non-firefighters (−33.38%∆ (95% CI: −54.97, −1.43)). Years of firefighting was not associated with a decrease in AMH. Firefighters in this study had lower AMH levels than non-firefighters. More research is needed to understand the mechanisms by which firefighting could reduce AMH and affect fertility.

## 1. Introduction

Environmental exposures can have adverse effects on a woman’s reproductive health [1,2,3]. Female firefighters have a variety of environmental and occupational exposures that can negatively impact their reproductive health by increasing their risk of miscarriage and pre-term birth, as well as increasing their use of fertility drugs [4,5,6,7,8,9,10,11].

Anti-müllerian hormone (AMH) is secreted by the granulosa cells of follicles and plays a role in follicular development [12]. AMH gradually increases until puberty and then generally plateaus until around age 25 [13]. After this, as a woman ages, her AMH steadily declines to mirror the decline in number of oocytes [14,15]. AMH is used as a clinical marker of ovarian reserve and is often used as a marker of responsiveness to fertility treatment [12]. However, AMH is a novel biological marker, so few studies have measured AMH on a population level [16]. AMH may be useful in measuring the effects of exposures that target the ovary, and reduction in AMH levels have been associated with inhaled environmental exposures including smoking, burning fuel indoors for cooking or heating, and the indoor spraying of pesticides for malaria control [17,18,19], and may be useful in measuring the effect of exposures that target the ovary [16]. Prior research in mice has shown that benzo[a]pyrene, a polycyclic aromatic hydrocarbon (PAH) [20], and PM2.5 exposure [21] were associated with decreased AMH. However, additional research is needed to better understand the influence of environmental and occupational exposures on AMH in human populations.

Firefighters are exposed to a number of occupational hazards that can poorly impact their health [22,23,24,25,26]. Therefore, the objective of this study was to investigate whether AMH levels were lower in female firefighters compared to civilian non-firefighting women and to investigate whether among female firefighters, there was a dose-dependent relationship between years of firefighting and AMH levels.

## 2. Materials and Methods

### 2.1. Study Design and Recruitment

Firefighter participants were primarily recruited from The Health and Wellness of Women Firefighters Study (HWWFS), an ongoing cohort study of female firefighters over the age of 18 in the United States and Canada [27]. The purpose of the HWWFS is to investigate the work environment, health, and perceived experiences of women firefighters in the fire service. The HWWFS recruited initial participants through multiple affinity group email lists that have been previously used, as well as snowball sampling, which allows current participants to recruit additional participants [4]. In addition, a small number of firefighters expressed interest in participating in our study after learning about it through an occupational health conference on women in the fire service and were invited to participate. Firefighters were asked to recruit one or more friends or relatives to serve as the comparison group. Participants received a USD 5 gift card incentive for their participation. All protocols and communications with research participants were pre-approved by the Institutional Review Board of the University of Arizona (Protocol #2007816290, Approved 29 June 2020).

All participants were asked to respond to an online survey regarding their occupational and health history. Women were excluded from the analyses if they had a history of chemotherapy or radiation, were current cigarette smokers, or had an oophorectomy (Figure 1; *n* = 19), as these have been shown to affect AMH levels [17,28,29]. We also excluded participants (*n* = 81) who did not provide a dried blood spot sample, had missing information on firefighting status, and one duplicate response. Lastly, we excluded participants who did not supply enough blood for measurement (*n* = 13). Participants with history of certain autoimmune diseases (polyglandular syndrome, lymphocytic oophoritis, Addison’s disease, Hashimoto thyroiditis, or Celiac disease) were also excluded (*n* = 3) as these conditions can diminish ovarian reserve [30,31,32,33,34,35].

### 2.2. Measures

#### 2.2.1. Exposure

The primary exposure of interest was history of firefighting based on self-report. Among current and past firefighters, we also investigated occupational exposures. Years in the fire service and the number of live fires responded to in a typical month were treated as continuous variables. Poor fit of personal protective equipment (PPE) was a binary variable defined as a Likert Scale Score less than 3 on a scale of 1 = “Does not fit” to 5 = “Fits very well” for any PPE category (Firefighting turnout coat/pants; Firefighting boots; Self-contained breathing apparatus (SCBA); Firefighting helmet; Firefighting hood (standard); Firefighting hood (vapor); Firefighting gloves; Work gloves; Eye/face protection other than SCBA face piece). The same method was used to create a variable for poor fit of SCBA.

#### 2.2.2. Outcome

All participants received an AMH testing kit (Ansh Labs, Webster, TX, USA) to collect dried blood spots (DBS). The collection of the DBS required the participants to prick their fingers using lancets and provide five DBS for measurement. After collection, the dried blood spots were returned to the University of Arizona and stored at −20 °C until all samples were collected. Once all samples were collected, they were sent to Ansh Labs/Motive Biosciences Inc. (Webster, TX, USA) to assay AMH using the picoAMH ELISA [36]. Motive Biosciences reported a coefficient of variation of 9.4–9.5% for this analysis. Participants were given the option to receive the results of their AMH assay.

AMH (ng/mL) was log-transformed for all analyses to account for its non-normal distribution. Log-transformed AMH (ng/mL) was analyzed as a continuous variable. AMH values below the limit of detection (LOD) (0.03 ng/mL) were recoded as 0.03/sqrt(2). AMH values above the upper limit of quantification (ULOQ) (>17.3 ng/mL) were recoded as 17.3 ng/mL.

#### 2.2.3. Other Covariates

Information on age, race (White, Non-White), ethnicity (Yes: Hispanic, Latina, or Spanish origin; or No), annual household income (less than USD 25,000; USD 25,000–50,000; USD 50,001–75,000; USD 75,001–100,000; or more than USD 100,000), highest educational attainment (high school graduate or GED; some college or technical school; college graduate; or advanced degree), marital status (Married; In a registered domestic partnership or civil union; Separated; Never married; or Divorced), and age at menarche (≤11; 12; 13; ≥14) were based on self-reported data at enrollment into our sub-study. Participants were asked about their typical level of exercise. Exercise habits ranged from 1 to 5 (1 indicating sedentary activity and 5 indicating very strenuous exercise). Current exercise habits were calculated using the level of exercise they indicated for their current age group. To determine the burden of infertility, participants were asked if they ever tried to become pregnant for more than one year without success. Participants also reported if they ever received fertility treatment. Information on other reproductive conditions was also collected, such as endometriosis (Yes, laparoscopically confirmed; Yes, not laparoscopically confirmed; No), history of hormonal contraceptive use (ever and current), and current contraceptive method (Surgery (sterilization), birth control pill (oral contraceptives), male condom, other hormonal methods (implant, injectable, contraceptive patch, contraceptive ring), intrauterine device (IUD), periodic abstinence, other, none of the above). Hormonal contraceptive use was analyzed categorically (birth control pill; other hormonal methods; or neither). Body mass index (BMI) was calculated based on self-reported height and weight and converted to kg/m^2^. BMI was reported as a continuous variable and categorized (BMI < 18.5; 18.5 ≤ BMI < 25; 25 ≤ BMI < 30; BMI ≥ 30). History of other autoimmune diseases was self-reported as a categorical variable. Perceived stress was measured using the 10 Question Perceived Stress Scale (PSS-10), with a higher score indicating a higher perceived stress level [37]. Participants also reported the number of known pregnancies they have had, including both miscarriages and births, as well as the number of biological children they have. Two participants reported having 14 biological children despite only having four pregnancies, so we chose to omit these two responses from the number of biological children.

### 2.3. Statistical Methods

T-tests were performed using the Satterthwaite method to produce *p*-values for continuous variables in in Table 1 and Chi-Squared tests were performed for categorical variables. Frequencies < 5 were suppressed in Table 1 and Table 2 to ensure participant privacy.

Linear regression models were utilized to estimate the association between firefighting status, firefighting exposures, and continuous, log transformed AMH. All models were adjusted for age and age^2^, to account for a non-linear association between age and AMH, and BMI. These confounding factors were determined based on *a priori* associations in the published literature. Percent changes in AMH between exposure groups were calculated from the estimated regression coefficients (β) as ([exp(β) − 1] ×100) and presented with corresponding 95% confidence intervals (CIs) (Table 3). We also investigated effect modification by age, by stratifying results by age group (<32, 32–37, and 38+) and conducted likelihood ratio tests comparing a model with an interaction term between age categories and exposure and a model without.

We restricted our analyses to non-smokers, as lowered AMH has been associated with smoking [3,18]. As women with polycystic ovary syndrome (PCOS) are known to have higher age-standardized AMH levels [38,39], in sensitivity analyses we excluded participants with probable PCOS (*n* = 20), defined as either having self-reported PCOS and/or an AMH value above 10 ng/mL [40].

## 3. Results

We included 106 firefighters and 58 non-firefighters in our main analysis (Figure 1). Firefighters and non-firefighters had mean ages of 38 and 35 years, respectively (*p*-value: 0.02) (Table 1). In total, 44.8% of firefighters had a BMI between 25 and 30 kg/m^2^ and 15.2% had a BMI greater than 30 kg/m^2^, while 26.3% of non-firefighters had a BMI between 25 and 30 kg/m^2^ and 19.3% had a BMI greater than 30 kg/m^2^. Firefighters indicated a higher level of perceived stress compared to non-firefighters (*p*-value: 0.04). Firefighters reported having fewer pregnancies and biological children than non-firefighters; however, these findings were not statistically significantly different. Participants were overwhelmingly white (95.7%), which is congruent with demographics of past studies on this population and the fire service in general [4]. More firefighters reported very strenuous exercise habits than non-firefighters, while more non-firefighters reported sedentary habits (*p*-value: 0.001). Our findings suggest that a larger proportion of firefighters reported an early age at menarche (age < 13) [41], history of infertility, and PCOS than non-firefighters; however, these results were not statistically significantly different between groups.

Among participants who were firefighters, most were currently working in that occupation at the time of the study (91.5%) (Table 2). The average length of years in the fire service was 13.3, ranging from 2 to 34 years, and the average number of calls responded to in a typical month was 62, with 2 of those calls being live fires. Within the fire service, participants mainly worked as firefighters, drivers, or operators with the next most common ranks being firefighter/paramedic and company officer (Lieutenant/Captain). Over a third of firefighters participated in wildland firefighting, either part-time or full-time. Over half of firefighters reported poor overall fit of at least part of their PPE, with approximately one quarter of firefighters reporting poor fit of their SCBA.

Overall, firefighters had a lower mean AMH compared to non-firefighters in crude models and this association remained statistically significant after adjustment for potential confounding factors (−33.4%∆ (95% CI: −55.0, −1.4)) (Table 3). Results from sensitivity analyses excluding participants with probable PCOS generally supported the primary analysis (Appendix A).

For analyses investigating occupational factors among firefighters, we observed a 54.6% lower AMH for every 5 years worked in the fire service (95% CI: −63.4, −43.7) (Table 4). However, this attenuated after adjustment for potential confounding factors, age, and BMI (0.70%∆; 95% CI: −25.11, 35.43). We observed no change in AMH for number of live fires responded to in a typical month, for poor fit of at least one form of the PPE, for poor fit of their SCBA, or for wildland firefighting. We observed similar patterns in sensitivity analyses excluding participants with diagnosed or suspected PCOS (Appendix A). Results stratified by age group showed that firefighters 38 years or older had a 63.5% lower AMH than non-firefighters in that age group (95% CI: −83.8, −17.9) (Appendix A). However, there was no statistically significant difference in firefighter status and AMH between age groups (interaction *p*-value = 0.71) (Appendix A).

## 4. Discussion

Women firefighters may be at greater risk for adverse reproductive outcomes compared to a civilian population; however, there has been limited research on this topic. We observed that female firefighters had lower levels of AMH compared to civilian women, a marker of ovarian reserve and reproductive function. These findings, along with prior research, suggest that more research on the reproductive health and fertility of female firefighters is warranted [4].

In our main analyses, we observed that firefighters had a 33% lower mean AMH level compared to non-firefighters after adjustment for confounders. These findings support evidence from previous studies that examined the effect of other environmental exposures on AMH levels. A prior cross-sectional analysis found that current smokers had a 44% lower AMH (95% CI: −68, −2) compared with never smokers [17]. In a 2016 analysis of the Sister Study, women who smoked 20 cigarettes per day had lower AMH compared to non-smokers (−56%, 95% CI: −80, −3) [18]. Given these and other findings on the association between smoking and time of menopause, we restricted our analyses to non-smokers. The Sister Study also observed that women who reported burning wood indoors had a 36% lower AMH (95% CI: −56, −8) compared to those who did not; and women who reported indoor use of artificial fire logs had a 46% lower AMH (95% CI: −67, −10) [18]. In an analysis of AMH levels of South African women, researchers found that women exposed to pyrethroids from indoor pesticide spraying for malaria control had a 25% lower AMH (95% CI: −39, −8) than those who were not exposed [19]. Given the potentially large clinical magnitude, the similarity with other findings, and the large uncertainty in the estimated effects, our results should be replicated using larger sample sizes.

We examined the relationship between AMH and continuous years in the fire service among firefighters to determine if there is an inverse dose–response relationship between number of years in the fire service and AMH levels. We hypothesized that women who have been firefighters longer would have had more exposure to inhaled environmental exposures, decreasing their AMH. When we examined this relationship, we observed a statistically significant decrease in AMH for every five years of service, but this attenuated after adjusting for age and BMI. This is most likely driven by the high correlation between years in the fire service and age.

We observed no association between number of live fires in a typical month, poor fit of PPE, poor fit of SCBA, or wildland firefighting and AMH levels. We had hypothesized that we would observe a decrease in AMH among firefighters who reported poor fit of the PPE and SCBA, because of the possibility for increased exposure to inhalants [1,2,3]. Over half of firefighters reported poor fit of at least one item of their PPE, and almost a quarter of firefighters reported poor fit of their SCBA. Previous research demonstrated that both male and females reported poor fit of their gear, but female firefighters may be more likely to select larger sized gear to better compensate for fit in the bust and hip, and these larger uniforms may lead to injury [42]. Further research is needed on the association between fit of PPE and AMH among female firefighters. The large confidence intervals presented are indicative of the small sample size, further supporting the need for research on this population with a larger sample size to allow for increased precision.

In addition to environmental exposures, there may be other exposures related to firefighting that may influence AMH levels. Prior research has found that psychological stress was negatively associated with serum AMH in infertile women [43]. Firefighters are subject to repeated exposure to trauma [44] and higher risk of depression than the general population [45]. Future studies should examine the relationship between stress and AMH, as well as other occupational factors, such as shift work [46]. Future studies investigating stress may also be enhanced by measuring stress hormones.

Inhalation and dermal exposure to smoke and PAH may reduce AMH [4,5,6,20], but the potential mechanism by which firefighting influences AMH is not fully understood. The mechanism likely involves multiple pathways that target follicular development possibly related to smoke inhalation and stress. More research is needed to investigate the pathway between firefighting and reduced AMH.

This study has many strengths including its novel focus on occupational exposures among female firefighters [4] and the utilization of AMH as a biological marker on a population level [16]. To our knowledge, this is one of the first studies to use this biomarker in occupational health research. However, there are important limitations that must be recognized. One of the biggest limitations of this study is its comparatively small sample size. Future studies should focus on recruiting firefighters at a local and national level via social media to increase participation, as recruiting through social media has previously been successful [47,48], and special attention should be paid to recruiting a representative comparison population. Participants from this study were recruited from a number of sources but women who are interested in contributing to reproductive health research because of their own adverse experiences may have been more likely to contribute. Additionally, there may be measurement error of AMH levels due to sample collection which involved participants collecting their own samples at home and returning in the mail, Prior research has suggested that AMH collected via DBS is stable across the menstrual cycle and up to two weeks at room temperature [49,50]; however, degradation and other unknown factors may have affected the quality of the biomarkers. For example, there was variation in time outside of the storage freezer across samples and some samples were sent to us without the required desiccant packet, which was added before storage. We would expect any misclassification of AMH to be nondifferential with respect to the exposure. In addition, all measures except AMH were self-reported, so their accuracy should be taken into consideration. However, we would expect that our primary exposure, history of working as a firefighter, will have been recalled accurately. Future studies should also differentiate between volunteer and career firefighters, as their level of exposures may differ and affect their risk of reproductive health outcomes [11]. Additionally, we did not collect detailed occupational information from non-firefighter participants. It is possible that occupational status of non-firefighters may also influence their AMH levels. However, we would expect this to attenuate any observed associations. Future studies should collect detailed information on occupation. There may be other covariates, such as diet and exercise which may serve as potential confounders. However, there is limited and conflicting existing literature on whether diet or physical activity influence AMH levels among the general population [51,52,53,54]. We collected information on history of infertility and utilization of fertility treatments, but we did not collect information on what influenced participants to pursue fertility treatment or not. There are many factors a person may consider when deciding whether or not to utilize fertility treatments including financial, health status, and lifestyle [55]. Given our small sample size, it is difficult to draw meaningful conclusions about the differences in reported infertility and utilization of fertility treatments in our study population. Future studies should collect more information about fertility treatments. Lastly, we removed participants with autoimmune conditions that are known to influence AMH, but some conditions may be undiagnosed. However, we would expect the influence to be minimal and the prevalence of undiagnosed individuals to be similar between groups.

## 5. Conclusions

There is increasing interest into the reproductive health of female firefighters. Our results suggest that there may be an association between exposures related to working as a firefighter and a biological marker of reproductive health, AMH. Future research should include a larger sample of female firefighters and non-firefighters as well as longitudinal analysis to further investigate these findings.

## Figures and Tables

**Figure 1 ijerph-19-05981-f001:**
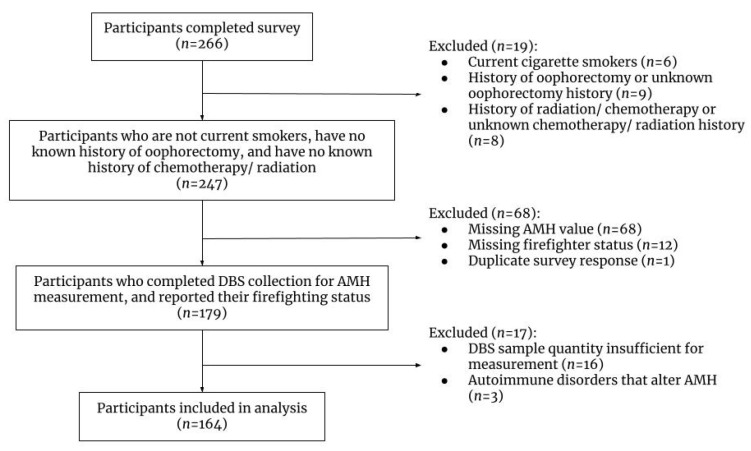
Participant flow diagram. Totals excluded do not sum to sub-categories, as some participants met multiple exclusion criteria.

**Table 1 ijerph-19-05981-t001:** Demographic characteristics of study sample stratified by history of employment as a firefighter (*n* = 164).

Characteristic	Firefighting Status, *n* (%)				*p*-Value
	Firefighter,		Non-Firefighter,		
	106 (64.6)		58 (35.4)		
	Mean ± SD, [Range]				
Age, years	38 ± 7.2, [20, 55]		35.3 ± 6.6, [20, 54]		0.02
Perceived stress ^a^	15.6 ± 6.9, [3, 34]		13.4 ± 6.1, [0, 27]		0.04
BMI, kg/m^2^	26.3 ± 4.5, [17.7, 42.9]		25.8 ± 5.2, [17.5, 38.7]		0.55
Number of known pregnancies	1.2 ± 1.5, [0, 6]		1.6 ± 1.8, [0, 6]		0.18
Number of biological children	0.8 ± 1.0, [0, 5]		1.2 ± 1.3, [0, 5]		0.15
	*n*^b^ (%)				
BMI, kg/m^2^					0.13
BMI < 18.5	<5		<5		
18.5 ≤ BMI < 25	41	(38.7)	30	(51.7)	
25 ≤ BMI < 30	47	(44.3)	15	(25.9)	
BMI ≥ 30	16	(15.1)	11	(19.0)	
Race					0.17
White	101	(95.3)	56	(96.6)	
Ethnicity					0.88
Hispanic	8	(7.5)	<5		
Annual household income					0.24
USD 50,000 or less	11	(10.4)	8	(13.8)	
USD 50,001–75,000	20	(18.9)	5	(8.8)	
USD 75,001–100,000	19	(17.9)	12	(21.1)	
More than USD 100,000	56	(52.8)	33	(56.1)	
Highest educational attainment					0.003
High school graduate or GED, some college, or technical school	46	(43.4)	11	(19.0)	
College graduate (4-year degree)	43	(40.6)	27	(46.6)	
Advanced degree (graduate or professional)	17	(16.0)	20	(34.5)	
Marital status					0.30
Married or in a registered domestic partnership or civil union	59	(55.7)	41	(66.7)	
Never married	27	(25.5)	12	(21.1)	
Divorced or separated	19	(17.9)	5	(7.0)	
Current exercise habits					0.001
1: Sedentary	<5		<5		
2	<5		6	(10.3)	
3	31	(29.2)	27	(39.7)	
4	45	(42.5)	19	(32.8)	
5: Very strenuous	27	(25.5)	<5		
Age at menarche					0.41
≤11	26	(24.5)	8	(13.8)	
12	28	(26.4)	17	(29.3)	
13	21	(19.8)	17	(29.3)	
≥14	31	(29.2)	16	(27.6)	
History of infertility	19	(17.9)	<5		0.05
Ever used fertility treatment	13	(12.3)	7	(12.1)	0.97
Ever used hormonal contraceptives					
Oral contraceptives	85	(80.2)	48	(82.8)	0.69
Other hormonal contraceptive (implant, injectable, patch, ring)	32	(30.2)	15	(25.9)	0.56
Currently using hormonal contraceptives					
Oral contraceptives	13	(12.3)	6	(10.3)	0.71
Other hormonal contraceptive (implant, injectable, patch, ring)	<5		<5		0.83
Endometriosis history	8	(7.5)	<5		0.58
History of polycystic ovary syndrome	6	(5.6)	<5		0.53

^a^ Measured with the PSS-10 (10 Question Perceived Stress Scale). Cells with less than five participants have been censored to ensure participant privacy. ^b^ Cells with less than five participants have been censored to ensure participant privacy.

**Table 2 ijerph-19-05981-t002:** Occupational history restricted to female firefighters in our sample (*n* = 106).

Characteristic	Firefighters	
	(*n* = 106)	
	mean ± SD, [range]	
Years in the fire service	13.3 ± 6.8, [2, 34]	
Calls responded to in a typical month	61.7 ± 55.4, [0, 350]	
Live fires responded to in a typical month	2.3 ± 5.1, [0, 50]	
	*n*^c^ (%)	
Current or past firefighter		
Current	97	(91.5)
Past	9	(8.5)
Current rank ^a^		
Firefighter, Driver, or Operator	39	(36.8)
Firefighter/Paramedic	29	(27.4)
Company Officer (Lieutenant, Captain)	29	(27.4)
Chief (Battalion, Deputy, Other)	6	(5.7)
Other	<5	
Participation in wildland firefighting	37	(34.9)
Poor fit of at least one part of personal protective equipment ^b^	55	(51.9)
Poor fit of self-contained breathing apparatus (SCBA)	26	(24.5)

^a^ Past firefighters were asked for their highest achieved rank. ^b^ “Poor fit” defined as a Likert Scale Score Less than 3 where 1 = “Does not fit” and 5 = “Fits very well” on any personal protective equipment category (Firefighting turnout coat/pants; Firefighting boots; Self-contained breathing apparatus (SCBA); Firefighting helmet; Firefighting hood (standard); Firefighting hood (vapor); Firefighting gloves; Work gloves; Eye/face protection other than SCBA face piece). Cells with less than five participants have been censored to ensure participant privacy. ^c^ Cells with less than five participants have been censored to ensure participant privacy.

**Table 3 ijerph-19-05981-t003:** Percent change in anti-müllerian hormone levels comparing non-firefighters and firefighters (*n* = 164).

Firefighting Status	AMH (ng/mL), Mean ± SD, [Range]	Model 1 % Difference (95% CI)	Model 2 % Difference (95% CI)
Non-firefighters (*n* = 58)	4.37 ± 4.50, [0.16, 17.30]	0.00 (Ref.)	0.00 (Ref.)
Firefighters (*n* = 106)	2.93 ± 3.83, [0.02, 17.30]	−57.49 (−75.06, −27.54)	−33.38 (−54.97, −1.43)

Abbreviations: AMH—anti-müllerian hormone; CI—confidence interval; SD—standard deviation. Model 1: linear regression of log(AMH) and firefighting status. Model 2: adjusted for age, age^2^, and body mass index. % difference calculated by: ([exp(β) − 1] × 100).

**Table 4 ijerph-19-05981-t004:** Among firefighters, occupational exposures and percent change in anti-müllerian hormone (*n* = 106).

Firefighting-Related Exposure	AMH (ng/mL), Mean ± SD, [Range]	Model 1 % Difference (95% CI)	Model 2 % Difference (95% CI)
Years in the fire service continuous (per 5-year increase)		−54.60 (−63.39, −43.69)	0.70 (−25.11, 35.43)
Live fires responded to in a typical month		1.00 (−5.88, 8.37)	1.42 (−3.40, 6.48)
Poor fit of any personal protective equipment			
No (*n* = 51)	3.29 ± 4.54, [0.02, 17.3]	0.00 (Ref.)	0.00 (Ref.)
Yes (*n* = 55)	2.60 ± 3.05, [0.02, 15.1]	9.61 (−46.23, 123.46)	46.66 (−10.10, 139.25)
Poor fit of self-contained breathing apparatus (SCBA)			
No (*n* = 80)	2.99 ± 4.09, [0.02, 17.3]	0.00 (Ref.)	0.00 (Ref.)
Yes (*n* = 26)	2.75 ± 2.96, [0.02, 11.3]	22.88 (−46.22, 180.80)	56.13 (−13.03, 180.29)
Wildland firefighting			
No (*n* = 68)	2.99 ± 3.78, [0.02, 17.3]	0.00 (Ref.)	0.00 (Ref.)
Yes (*n* = 37)	2.79 ± 4.02, [0.02, 16.7]	8.96 (−48.53, 130.65)	−7.90 (−45.05, 54.35)

Abbreviations: AMH—anti-müllerian hormone; CI—confidence interval; SD—standard deviation. Model 1: linear regression of log(AMH) and firefighting-related exposure. Model 2: adjusted for age, age^2^, and body mass index. % difference calculated by: ([exp(β) − 1] × 100).

## Data Availability

The data presented in this study are available on request from the corresponding author.

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
