# Peer review of "Anti-Müllerian Hormone Levels among Female Firefighters"

_ijerph, 2022, doi:10.3390/ijerph19105981_

Round 1

Reviewer 1 Report

  • This is a very well designed and executed study, although it has several limitations as outlined by the authors. Nevertheless, given the importance of the findings in stimulating further research in the area, I would recommend it to be published after attending the following comments.
  • The authors are expected to carry out a statistical test to show whether there was a difference in characteristics presented in Table 1 between the two groups.
  • There was no association between fit of PPE/SCBA with the level of the hormone. What is the basis for the recommendation made in this regard in the conclusions section?
  • No mention was made in the manuscript about ethical clearance of the research. Authors need to provide ethical clearance/no ethical concern exists in the study from an IRB. The incentive $5 given to participants can be considered as an inducement. Authors need to explain about the incentive. 

Reviewer 2 Report

In this Manuscript the authors describe Anti-Mullerian hormone (AMH) levels in firefighters and non-firefighter,s suggesting that the former have lower levels and this may be an occupational-related challenge in terms of infertility. It is an interesting study that has the usual issues related to similar ones performed in volunteer human populations and relying heavily on self-reporting. Although a very credible effort was made in order to focus the text (and notably the abstract) strictly on the data, there are a few issues the authors should address.

-The control used for non-firefighters were volunteers, but no occupational stratification is shown, including possible stressful or heavy exercise jobs. The authors should discuss this or show that data.

-The non-firefighters are in a much lower number. The authors should discuss how that may skew the data.

-An estimation of exercise should be included, and well as nutritional considerations, if available. If not, the potential involvement of these factors (known to affect female fertility) should be at least discussed.+

-It seems that several comparisons are not statistically significant, such as the higher stress levels in firefighters. This should be clearly stated in the text, when mentioned.

-While firefighters seem to have a higher reported history of infertility, this difference disappears when considering fertility treatments. The reason for this (financial, choice, etc) should be discussed, if available. More information on fertility treatments involved would also be very useful.

-Information on children (yes/no) is also lacking, and this seems odd given the purpose of the study.

-In the discussion the authors could include the possibility of monitoring stress hormones in this context for future studies.

Reviewer 3 Report

This is an interesting paper with a sound statistical analysis and an appropriate design. As biostatistician, I would only bring up some minor remarks. The authors could add to table 1 an additional column reporting the significance of the between group comparisons in order to better highlights potential confounding factors that could be used as adjusting factors in multivariable logistic models. Did the authors try to perform, due to the observational design, a propensity score weighting in order to properly estimate the effect of the exposure? Please add some details on such issue.  It is also not clear to me why the authors, in table 1, reported small frequencies as <5 and only for some variables. Please specify all the frequencies in an exact way. 

Round 2

Reviewer 2 Report

The authors have adequately addressed my previous concerns. I have no further comments.

Author Response

Thank you for your feedback.